# Investigating hill sheep farmers and crofters' experiences of blackloss in the Highlands and Islands of Scotland

Fiona McAuliffe[1,2]*, Ann McLaren[2], Neil Sargison[1], Franz Brülisauer[3], Andrew Kent[4], Davy McCracken[2]

**1** Royal (Dick) School of Veterinary Studies, University of Edinburgh, Midlothian, United Kingdom, **2** Hill & Mountain Research Centre, SRUC: Scotland's Rural College, Crianlarich, United Kingdom, **3** SRUC Veterinary Services, Inverness, United Kingdom, **4** NatureScot, Inverness, United Kingdom

* fiona.mcauliffe@sruc.ac.uk, F.Mc-Auliffe@sms.ed.ac.uk

**Data Availability Statement:** The data collected for the present study may contain details which could potentially lead to the identification of study participants. As such, requests for access to this

## Abstract

Hill sheep farming is an important component of Scottish agriculture and comprises a significant land use in much of the Highlands and Islands. However it faces significant challenges due to the natural constraints of the landscape. Hill sheep farming uses hardy traditional breeds, such as the Scottish blackface and North Country Cheviot to graze extensive areas, where the sheep are not housed and tend to lamb on the open hill. Flocks are gathered several times a year for stock checks, husbandry, and health treatments. Between these handling events, stock will disappear and be unaccounted for. These unexplained losses are known as blackloss in the Highlands and Islands. Previously reported figures for annual lamb blackloss give an average of 18.6%. These losses are in addition to the known losses of lambs and represent a significant welfare and sustainability issue. High parasite burdens, predation, a photosensitisation disease known as plochteach or yellowses, and poor nutrition are often given as presumed reasons for blackloss. A questionnaire was developed to assess the experiences, impacts and understanding flock managers have of blackloss. Typology analysis using partitioning around medoids was used to cluster respondents into three distinct groups: 1- very large extensive farms and Sheep Stock Clubs, 2- medium sized farms, and 3- small-scale crofts. The responses of these groups were subsequently analysed to see if their experiences and perceptions of blackloss differed with relation to lamb health challenges and predation impacts. The groups reported similar health challenges, apart from Group 1 which had a significantly higher plochteach challenge. In terms of predators, Group 1 also perceived white-tailed eagles (*Haliaeetus albicilla*) as a much higher threat to their lambs than the other groups. It was observed that many of the respondents believed blackloss is inevitable and that predators pose a large threat to lambs. However, most agreed that reducing these losses is important and that understanding the causes would enable them to do so.

data may be sent to SRUC's Data Protection Officer. Address: Scotland's Rural College, Executive Office, Peter Wilson Building, Kings Buildings, West Mains Road, Edinburgh EH9 3JG. Telephone: 0131 535 4432 E-mail: dpo@sruc.ac. uk.

**Funding:** This study was undertaken as part of a PhD studentship, funded by Scotland's Rural College (SRUC) and NatureScot [SRUC Studentship 1033123: Investigation of causes of lamb loss on Highland farms and crofts]. The PhD is registered at the University of Edinburgh's Royal (Dick) School of Veterinary Studies. It involves close collaboration between SRUC, The Royal (Dick) School of Veterinary Studies, NatureScot and SRUC Veterinary Services. The funders had no role in study design, data collection and analysis, decision to publish, or preparation of the manuscript.

**Competing interests:** Co-author AK is employed by the funding organization (NatureScot). This does not alter our adherence to PLOS ONE policies on sharing data and materials. All other authors declare that they have no competing interests.

## Introduction

Blackloss is the term used for the unexplained losses of lambs on extensive hill sheep systems in the Scotland. These losses are in addition to the known losses of lambs and represent a welfare and sustainability issue. The very nature of hill sheep farming in Scotland makes quantifying levels of blackloss and identifying potential causes quite challenging. Although the cause of death of blackloss lambs remains unexplained, high parasite burdens, predation, disease, and poor nutrition are often given as presumed reasons for the losses as these are known to affect other lambs within the cohort. A previous questionnaire study aimed to explore the level of losses experienced by a wide population of sheep farmers in the Highlands and Islands [1]. They received 40 responses from which they found an average marking to weaning, viable, lamb loss of 6.6%. The reasons given by respondents for these losses ranged from health issues such as tick bourne diseases, plochteach/yellowses and 'braxy' (*Clostridium septicum*), to predation by foxes (*Vulpes vulpes*), ravens (*Corvus corax*), hooded crows (*Corvus cornix*) and eagles (*Aquila chrysaetos* and *Haliaeetus albicilla)*, and finally accidents such as open drains [1]. Whilst this study provided a much-needed insight into the levels of blackloss experienced in hill sheep systems and hinted at some of the causes of the losses, it did not elaborate beyond these anecdotal causes. Therefore, to investigate this further, a questionnaire was developed to assess the experiences, impacts and understanding hill sheep farmers and crofters have of blackloss.

Sheep flocks each tend to have their own unique set of resources, management schemes and challenges facing them [2]. In order to study how blackloss might be impacting the flocks of questionnaire respondents, it would be valuable to classify the respondents into different groups, based on variables related to their hill systems. Questionnaire data are often analysed using a multivariate-based typological approach to group respondents into clusters. Typology analysis is a way of describing groups of respondents displaying different clusters of behaviours, attitudes or views of the world, and is a system used for putting things into groups according to how they are similar. This approach, using cluster analysis, allows for complex and varied livestock systems to be classified into broadly similar groups, and has been used for sheep systems in the past [2–4]. This technique helps to identify trends in datasets with a broad range of categories, variables or apparently disassociated factors which might otherwise be unclear. Clusters help us to better understand the many attributes that may be associated with blackloss, whether there are distinct clusters of respondents that may suffer disproportionately from blackloss in comparison to other groups, and if there might be a driving reason for this, e.g. a higher proportion of their lambs were affected by a given health issue or the grouping perceived a higher risk of predation etc.

Partitioning Around Medoids (PAM) clustering, also known as *k*-medoids clustering [5], follows a very similar technique to *k*-means clustering [6], however rather than finding the centroid of the cluster this technique finds the most representative object within the cluster, known as the medoid [5]. After finding a set of *K* representative objects, the *K* clusters are constructed by assigning each object of the dataset to the nearest representative object. PAM clustering has been used to group farms in regard to dairy cow production parameters and bulk tank milk antibody status of internal parasites [7], classifying German farmers for policy design [8], and to determine bacterial communities associated with methane emissions in sheep [9].

In order to investigate blackloss within the Highlands and Islands and ultimately reduce future losses by providing the basis for change, it is important to gather knowledge and experiences of those affected using a questionnaire; the farmers, crofters and flock managers involved in the hill sheep industry of the region. The principal objective of this questionnaire was to

understand the main causes (perceived or real) and consequences of blackloss on their sheep enterprise, lamb health problems they have experienced within their flock as well as the potential roles predators might be playing in the losses of lambs. The study aimed to undertake a typological analysis to cluster respondents into distinct groups, and to subsequently determine whether these groups differ significantly with regards to their experience of and views on blackloss.

## Methodology

### Questionnaire design

A questionnaire was developed to investigate hill sheep farmer and crofters' management practices and experiences of blackloss in lambs. The questionnaire consisted of 32 questions and followed guidelines set out in questionnaire design textbooks [10–12]. The questions posed covered aspects about the sheep system type, its size and land cover, access to equipment and infrastructure, sheep numbers and breed, flock management and husbandry practices, levels of blackloss experienced, health challenges facing their lambs, predator presence and their impact on lambs, and the respondent's attitudes towards blackloss (see SI 1). The questionnaire went through several rounds of review by sector specialists to ensure the questions posed were relevant and appropriately phrased. Question formats varied and included yes or no responses (e.g. Do you condition score your ewes?), numerical (e.g. Number of breeding ewes), tick-box (Where are the majority of your single and twin lambs grazed between marking and weaning?), open response (e.g. In your opinion, what is 'blackloss'?), multiple choice (What predator species are present on your farm/croft?), or Likert scale response format (e.g. Blackloss is an inevitable part of hill sheep systems: 1 = Strongly disagree to 5 = Strongly agree). The survey was carried out by either digital or postal questionnaire, based on the preference of the participant.

### Ethics statement

The Human Ethical Review Committee (HERC) of The Royal (Dick) College of Veterinary Studies, University of Edinburgh granted approval for the questionnaire on the 21[st] of March 2020. Ethical approval was valid for the duration of the project, including activities related to the recruitment of participants to complete the questionnaire between 21[st] of March 2020 and 21[st] of March 2021. Participants were informed of their right to remain anonymous and about how their data would be stored and used in a General Data Protection Regulation (GDPR) statement at the start of the questionnaire. Participants then signed a written consent to take part in the study (see SI 1).

### Questionnaire distribution

Relevant organisations including SAC Consulting, SRUC Veterinary Services, the Scottish Crofting Federation (SCF), National Sheep Association (NSA), National Farmers Union of Scotland (NFUS) regional managers for Argyll & the Islands, the Highlands, and Forth & Clyde, and NatureScot's Sea Eagle Management Scheme (SEMS), were requested to make their members aware of the questionnaire and of how to obtain a copy. Hill sheep enterprises interested in taking part in the study would contact the researcher directly to request a copy of the questionnaire. The first questionnaire was dispatched on the 23[rd] of April, whilst the final one was sent on the 24[th] of December 2020.

Returned questionnaires were anonymised during analysis to preserve participant confidentiality. In total, there were 57 requests for a copy of the questionnaire, of which responses

were received from 31 participants, a response rate of 54.4%. Despite the impact of Covid-19 restrictions on distribution, the methods used still managed to reach a large proportion of the hill sheep farmers and crofters spread throughout the target Highlands and Islands region, and western Scotland more generally.

## Analysis

**Data preparation.** Returned questionnaires were anonymised and digitised using Microsoft Excel. Responses in Likert scale formats were assigned a value (e.g., Severe = 3, Mild = 2, None = 1, and Unknown = 0). A dataset containing numeric responses (from quantitative and discrete questionnaire items) was compiled for analysis. Questionnaire items were grouped into the following classifications: System, Husbandry, Blackloss, Lamb Health, Predation and Attitudes.

**Blackloss.** A total of twenty-three respondents felt that they suffered from blackloss, and of these thirteen returned lamb counts for marking and weaning during 2019. Following the methods used by [1], the average marking to weaning blackloss was calculated. The total number of all lambs weaned in the thirteen flocks was subtracted from the number of lambs which were marked, and then dividing by the total number of lambs that were marked to find the percentage blackloss. Following typology analysis, the percentage marking to weaning blackloss was calculated on a per cluster basis.

**Typology analysis.** Typology analysis was carried out by a series of steps using R and R Studio (Version 4.2.2) [13]. Variables relating to System (type, size, access to common grazing and the number of breeding ewes) were used in the clustering analysis. Clustering aims to determine how similar (or dissimilar) cases in a dataset are to one another so that they can then be grouped together. Each respondent was assigned a score based on the four variables selected, which was then used to determine the difference between respondents. The Gower Distance [14] metric was used to determine these scores, as this accounts for both continuous and discrete variables. To identify the optimal number of clusters with the largest silhouette width, the cluster analysis was performed with differing numbers of clusters (from 2 to 10). Ideally, the silhouette width should be a value of at least 0.25, and is a measure of how similar (+1) or dissimilar (-1) each case is to its assigned cluster [5, 15]. A larger silhouette width indicates high levels of clear cluster assignment [16]. Using this method, three clusters generated the highest average silhouette width of >0.7 and were used in the analysis. The Partitioning Around Medoids (PAM) method was then used for clustering, which uses observations from the raw dataset to define cluster centers (i.e. the four variables selected) [15]. A dendrogram heatmap was created using the clustering output, separating each respondent into their assigned clusters. ANOVA and Kruskal Wallis tests were used to verify that the clusters identified were significantly distinct.

Data on the experiences of respondents to other variables, including about their holding, husbandry practices, lamb health issues, predation, and attitudes towards blackloss, were not included in the main clustering analysis. This additional set of variables was used to further explore whether the clusters differed in how they manage their flocks, their experiences of blackloss, lamb health issues and predation challenges. Using R, ANOVA tests were used for continuous variables whilst Kruskal-Wallis (KW) tests were used for discrete variables to assess the statistical differences between the obtained cluster groups (similar to [2–4]). Post-hoc Tukey's HSD (honestly significant difference) and Dunn's tests for the ANOVA and Kruskal-Wallis analyses, respectively, were used to establish significant differences between each cluster.

## Results

### Typology analysis

The three clusters identified are described below and illustrated in Fig 1.

Group 1: This cluster is made up of two farms and three sheep stock clubs (SSC), with all bar one farm have access to common grazing. The holdings in this group are the largest with a median size of 2820 Ha (2672–3200 Ha) and 1000 breeding ewes (783–1438). The medoid of this cluster is respondent 11, a SSC of 2820 Ha, 1400 ewes and access to common grazing. This cluster can be considered the 'traditional extensive' group.

Group 2: This cluster comprises ten crofts and one SSC, which all have access to common grazing. They are the smallest in size, with a median of 45 Ha (12–600 Ha) and 120 breeding ewes (22–650). The medoid of this group is respondent 13, a croft of 45 Ha, 90 ewes and access to common grazing. This cluster can be considered the 'smallholders' group.

Group 3: This cluster includes thirteen farms and two crofts, with none having access to common grazing. They have a median size of 530 Ha (14–1500 Ha) and 500 ewes (17–943). This group's medoid is respondent 23, a farm of 955 Ha, 620 ewes and without access to common grazing. This cluster could be known as the 'medium enterprises' group.

The groups were significantly different from one another with relation to type (KW, $\chi^2(2) = 16.42$, p <0.001), size (ANOVA, $F_{2,28} = 105.8$, p <0.001), access to common grazing (KW, $\chi^2(2) = 26.90$, p <0.001), and number of breeding ewes (ANOVA, $F_{2,28} = 16.06$, p <0.001).

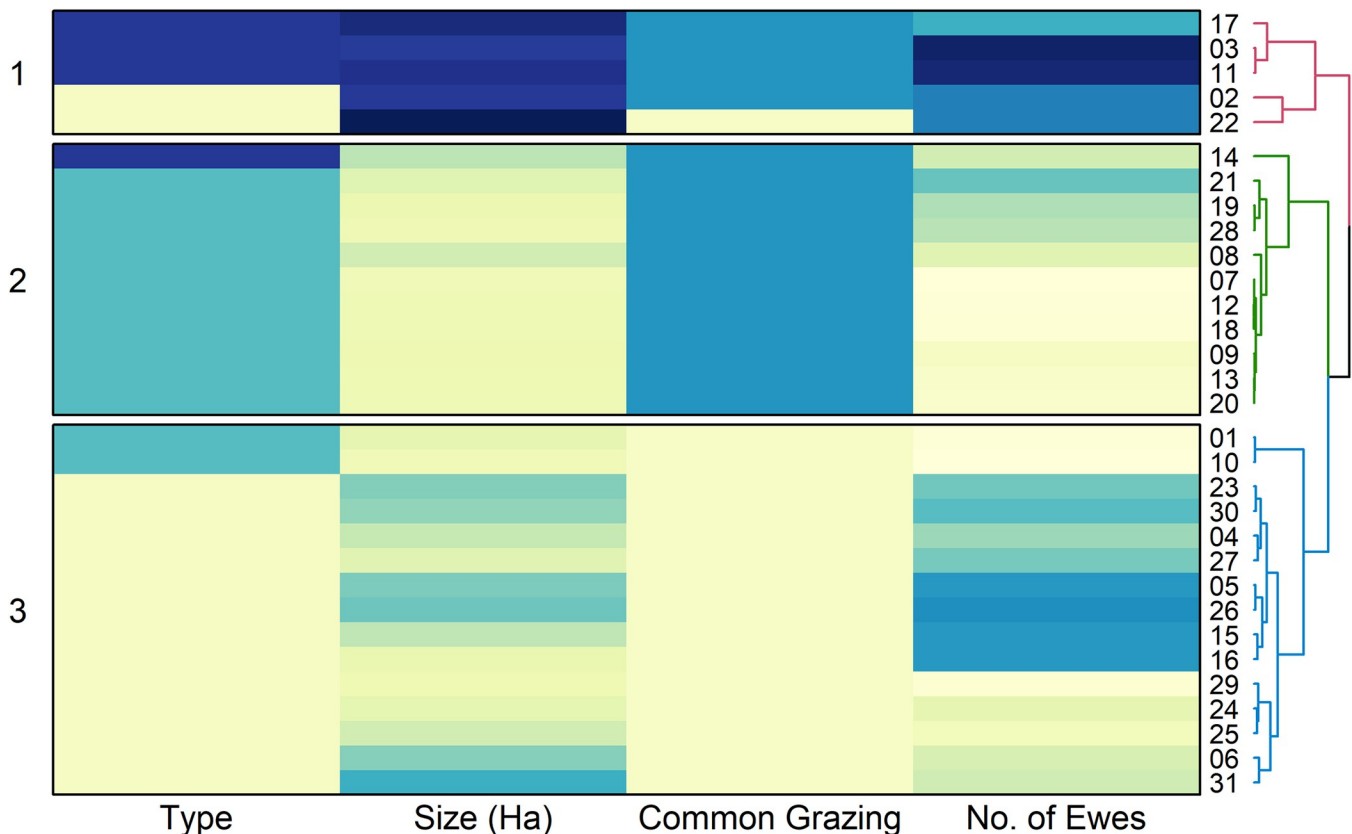

**Fig 1. Heatmap based on partitioning around medoids cluster analysis.** Partitioning around medoids clustering using Gower's distance was visualised in a heatmap to illustrate which cluster each of the respondents (01–31) sit within. Colours are assigned to scaled datum values, with darker blue indicating higher values (i.e. larger size Ha, more ewes). Type: Farms = yellow, crofts = light blue, SSC = dark blue. Common grazing access: no = yellow, yes = blue. The tree clusters and their shorter Gower distance indicate higher similarities. Similarity between respondents is represented by branch height; therefore, the lower a node is vertically, the more similar its subtree.

The clusters were each spread geographically, with Group 1 respondents located on the Isle of Skye, Kyle, and the Isle of Mull, Group 2 had participants from Lairg, the Isle of Skye and the Isle of Lewis, and finally Group 3 had the widest distribution, with respondents spread from the Isle of Arran in the south to Shetland in the north (Fig 2).

## Blackloss

The study found an overall marking to weaning blackloss, on an animal basis, of 7.6%. On a flock basis, average recorded blackloss levels within Group 1 were 1.3%, within Group 2 were 4.0% (0–5.8%), and were 13.5% (0–17.5%) within Group 3 (Table 1). However, the small sample sizes should be considered as only one respondent from Group 1 returned lamb counts. The average flock level loss was 6.3%.

## Cluster differences

**Husbandry.** The Groups did not differ in whether they condition scored or scanned their ewes; however, the scanning percentages were significantly different between groups (ANOVA, $F_{2,12}$ = 5.183, p = 0.024, Table 2). Group 3 had a significantly lower scanning percentage (114.8%), than either Group 1 or 2 (147.0 and 146.7% respectively). There was not a significant difference between groups in how old the majority of lambs are when they are marked or weaned, although Group 3 were the earliest marking (4.5 weeks) and Group 2 were the latest to wean their lambs (17.2 weeks). The study also found no difference in when the groups tag the majority of their lambs, with Groups 1 and 3 tagging at weaning, whilst Group 2 tag when the lambs are nine months old or when they leave the farm. Group 1 tended to rear both their single and twin lambs on open unimproved hill grazings between marking and weaning, whilst Group 2 also kept their singles in unimproved grazing areas they kept their twins on improved in-bye grazing, and finally Group 3 kept their singles in semi-improved parks and their twins on a mix of semi-improved and improved grazing areas, although these differences were not significant (Table 2).

**Blackloss.** The difference between groups in whether they suffered from blackloss was not significant, with all members of Group 1, 63.6% of Group 2 and 73.3% of Group 3 suffering at least some level of blackloss. When asked about the proportion of lambs lost to blackloss during the marking to weaning period for 2017, 2018 and 2019, Group 1 respondents reported consistently higher losses of 11–20% than either Group 2 or 3 (1–10%), although this difference was not significant (Table 3). One member of Group 1 reported very high losses >30% each year. When asked about what they considered to be the main causes of blackloss within their flocks, Group 1 members scored plochteach/yellowses as significantly more important than Group 2 (KW, $\chi^2(2)$ = 7.3, p = 0.02). Group 1 also scored hypothermia/exposure, parasites, trace element deficiencies, accidents, and theft as more important causes than Group 2 or 3, although these were not significant differences. Group 3 considered mismothering as a more important cause than Groups 1 or 2. All groups considered predators to be an important cause of blackloss. With regards to the consequences of blackloss, Group 1 felt that productivity loss was a more severe consequence than either Group 2 or Group 3 (KW, $\chi^2(2)$ = 8.15, p = 0.02). Group 1 also scored the financial loss as severe compared to Group 2 and Group 3, who felt it was a mild consequence (KW, $\chi^2(2)$ = 8.49, p = 0.01). Respondents within Group 1 scored their own stress as severe, whilst Group 2 felt it was a mild consequence (KW, $\chi^2(2)$ = 6.63, p = 0.03). Although not significantly different, Group 1 felt the impact on animal welfare and the loss of breeding potential were both severe consequences of blackloss, whilst Groups 2 and 3 felt they were mild. Finally, Group 1 felt that poor sustainability was a severe

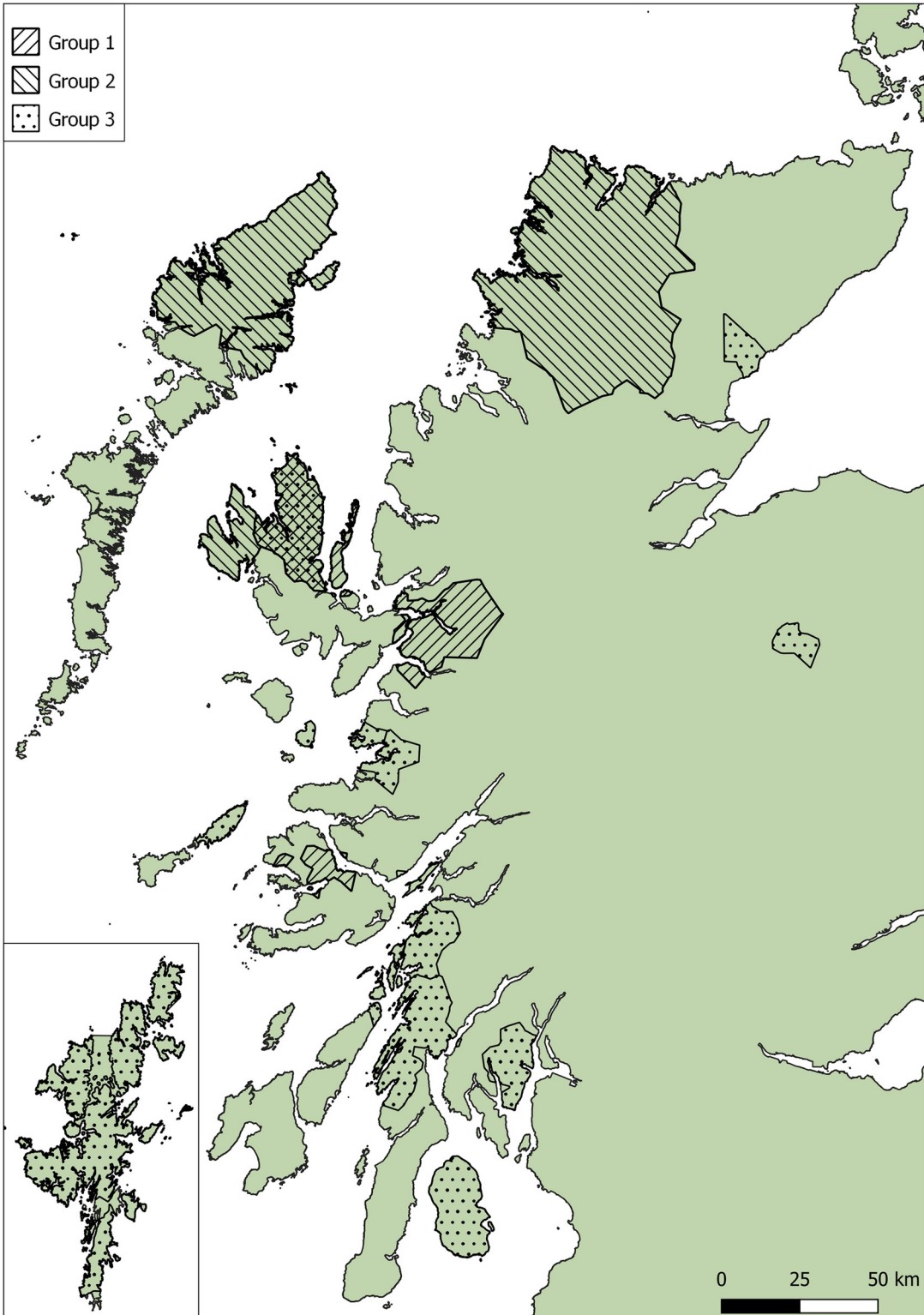

**Fig 2. Distribution of questionnaire respondents by their clusters.** Group 1- traditional extensive, Group 2- smallholdings, Group 3- medium enterprises. There was little overlap in where respondents were from, apart from the Uig area on the Isle of Skye, where all three clusters were represented (Inset Shetland). Figure was created using QGIS under a CC BY license, with permission from www.opendoorlogistics.com, original copyright Royal Mail data © Royal Mail copyright and database right 2015, and National Statistics data © Crown copyright and database right 2015.

**Table 1. Marking to weaning levels of blackloss (unexplained lamb losses).**

| Cluster | Number of flocks with returned marking and weaning lamb counts | Viable lamb loss (mark-to-wean) at flock level* % |
|---|---|---|
| | | Mean |
| | | Min-Max |
| Group 1 | 1 | 1.3 |
| | | na |
| Group 2 | 5 | 4.0 |
| | | 0–5.8 |
| Group 3 | 7 | 13.5 |
| | | 0–17.5 |
| Overall | 13 | 6.3 |
| | | 0–17.5 |

*The per cent of (the number of lambs in an individual flock that were marked minus the number of lambs in that flock that were weaned) divided by the number of lambs in that flock that were marked

consequence, Group 2 felt it was mild, and Group 3 felt it was not a consequence of blackloss (Table 3).

**Lamb health.** There were no significant differences between groups in the proportion of lambs which suffer from several health issues (trace element deficiencies (KW, $\chi^2(2) = 0.42$, p = 0.81), fluke (KW, $\chi^2(2) = 1.09$, p = 0.58), gastrointestinal worms (KW, $\chi^2(2) = 0.69$, p = 0.71), or tick borne diseases (KW, $\chi^2(2) = 1.53$, p = 0.47)). However, respondents in Group 1 reported a significantly higher proportion (1–20%) of their lambs suffer from plochteach/yellowses than Group 2 (KW, $\chi^2(2) = 7.30$, p = 0.02, Table 4).

**Predators.** The analysis did not find a significant difference between Groups in the reported impacts that most predators have on lambs that are less than 10 days old, apart from Group 1 which perceived white-tailed eagles to have a significantly higher impact than respondents in Group 3 who felt they had no impact (KW, $\chi^2(2) = 7.47$, p = 0.02, Table 5). The study found similar results when respondents were asked about the impact of predators on lambs

**Table 2. Questionnaire items relating to the flock husbandry and management practices on the holding.** The mean or median response (dependent on the test listed) for each of the Groups 1–3 is given.

| Question | Unit | Item | Test | 1 | 2 | 3 | Chi-square/ F-stat | p-value |
|---|---|---|---|---|---|---|---|---|
| Do you condition score your ewes? | 0 = No | | KW | 0 | 0 | 1 | 1.72 | 0.42 |
| | 1 = Yes | | | | | | | |
| Do you scan your ewes? | 0 = No | | KW | 0 | 1 | 0 | 2.57 | 0.28 |
| | 1 = Yes | | | | | | | |
| What is your average scanning percentage | % | | AN | 147.0 | 146.7 | 114.8 | 5.183 | 0.02 |
| How old are your lambs when they are: | Weeks | Marked | AN | 5.2 | 5.7 | 4.5 | 0.168 | 0.85 |
| | | Weaned | AN | 16.2 | 17.2 | 16.5 | 0.263 | 0.77 |
| When do you tag your lambs? | 1 = At lambing | | KW | 3 | 4 | 3 | 4.05 | 0.13 |
| | 2 = At marking | | | | | | | |
| | 3 = At weaning | | | | | | | |
| | 4 = When they leave the farm/ are 9 months old | | | | | | | |
| Where are your lambs between marking and weaning? | 1 = Improved grazing, 2 = Semi-improved grazing, 3 = Unimproved grazing | Singles | KW | 3 | 3 | 2 | 3.60 | 0.17 |
| | | Twins | KW | 3 | 1 | 1.5 | 4.99 | 0.08 |

**Table 3. Questionnaire items relating to the blackloss levels, causes and consequences on the holding.** The median response for each of the Groups 1–3 is given.

| Question | Unit | Item | Test | 1 | 2 | 3 | Chi-square | p-value |
|---|---|---|---|---|---|---|---|---|
| **Do you suffer from blackloss?** | *0 = No* | *Yes/No* | KW | 1 | 1 | 1 | 2.31 | 0.32 |
| | *1 = Yes* | | | | | | | |
| **The proportion of lambs you have lost to blackloss during the marking to weaning period** | *0 = Unknown, 1 = 0%,* | *2017* | KW | 3 | 2 | 2 | 3.44 | 0.18 |
| | *2 = 1–10%, 3 = 11–20%, 4 = 21–30%, 5 = >30%* | *2018* | KW | 3 | 2 | 2 | 5.95 | 0.05 |
| | | *2019* | KW | 3 | 2 | 2 | 3.91 | 0.14 |
| **The most important causes of blackloss of lambs on your holding?** | *0 = Unknown, 1 = Not important,* | *Mismothering* | KW | 1 | 0 | 2 | 3.54 | 0.17 |
| | | *Hypothermia/ Exposure* | KW | 2 | 0 | 1 | 1.48 | 0.05 |
| | | *Starvation* | KW | 0 | 1 | 1 | 1.47 | 0.05 |
| | *2 = Slightly important,* | *Predators* | KW | 3 | 3 | 3 | 3.29 | 0.19 |
| | *3 = Important* | *Parasites (worms, fluke, ticks)* | KW | 3 | 1 | 2 | 2.69 | 0.26 |
| | | *Plochteach/ Yellowses* | KW | **2** | **1** | 1 | 7.30 | 0.03 |
| | | *Trace element deficiencies* | KW | 2 | 1 | 1 | 3.95 | 0.14 |
| | | *Accidents* | KW | 2 | 1 | 1 | 3.69 | 0.16 |
| | | *Theft* | KW | 1 | 0 | 0 | 1.52 | 0.47 |
| **The main consequences of blackloss on your holding?** | *0 = Unknown, 1 = None,* | *Loss of productivity* | KW | **3** | **2** | **2** | 8.15 | 0.02 |
| | *2 = Mild,* | *Farmer/crofter stress* | KW | **3** | **2** | 2 | 6.63 | 0.04 |
| | *3 = Severe* | *Impact on animal welfare* | KW | 3 | 2 | 2 | 2.57 | 0.28 |
| | | *Financial loss* | KW | **3** | **2** | **2** | 8.49 | 0.01 |
| | | *Loss of breeding potential* | KW | 3 | 2 | 2 | 5.98 | 0.05 |
| | | *Poor sustainability* | KW | 3 | 2 | 1 | 4.95 | 0.08 |

older than 10 days, with respondent groups only differing significantly in opinion in relation to white-tailed eagles (WTE) and golden eagles (GE) (KW, $\chi^2(2) = 7.26$, p = 0.03). Group 1 again felt WTE have a high impact, whilst Group 3 felt the impact was low (KW, $\chi^2(2) = 10.75$, p<0.001). Group 1 felt GE's have a low impact, Group 2 thought they had no impact and Group 3 stated the impact was not applicable, as they tend to not occur on their holdings.

**Attitudes.** When asked the extent to which they agreed or disagreed with the following statements, no significant differences were found in how the Groups responded to three of the statements (Table 6). Groups 2 and 3 agreed although Group 1 neither agreed nor disagreed that '*blackloss is an inevitable part of hill sheep systems*'. Group 1 strongly agreed whilst Groups

**Table 4. Questionnaire items relating to the lamb health issues relevant to blackloss.** The median response for each of the Groups 1–3 is given.

| Question | Unit | Item | Test | 1 | 2 | 3 | Chi-square | p-value |
|---|---|---|---|---|---|---|---|---|
| **What percentage of your lambs suffer from the following diseases in an average year?** | *0 = Unknown* | *Plochteach/Yellowses* | KW | **2** | **1** | 1 | 7.30 | 0.03 |
| | *1 = 0%* | | | | | | | |
| | *2 = 1–20%* | *Trace element deficiencies* | KW | 2 | 1 | 1 | 0.42 | 0.81 |
| | *3 = 21–40%* | | | | | | | |
| | *4 = 41–60%* | *Fluke* | KW | 2 | 1 | 1 | 1.09 | 0.58 |
| | *5 = 61–80%* | | | | | | | |
| | *6 = 81–100%* | *Gastrointestinal worms* | KW | 2 | 2 | 2 | 0.69 | 0.71 |
| | | *Tick fever/ tick pyaemia/ louping ill* | KW | 2 | 1 | 2 | 1.53 | 0.47 |

**Table 5. Questionnaire items relating to the impact of predators on both young and older lambs.** The median response for each of the Groups 1–3 is given.

| Question | Unit | Item | Test | 1 | 2 | 3 | Chi-square/ F-stat | p-value |
|---|---|---|---|---|---|---|---|---|
| **Impact predators have on lambs which are less than 10 days old** | *0 = NA* | *Foxes* | KW | 1 | 3 | 3 | 2.37 | 0.31 |
| | *1 = None* | *Badgers* | KW | 0 | 0 | 1 | 5.72 | 0.06 |
| | *2 = Low* | | | | | | | |
| | *3 = Medium* | | | | | | | |
| | *4 = High* | *Ravens* | KW | 3 | 4 | 3 | 0.40 | 0.82 |
| | | *Crows* | KW | 3 | 3 | 2 | 5.23 | 0.07 |
| | | *Black-backed gulls* | KW | 2 | 3 | 2 | 0.89 | 0.64 |
| | | *White tailed eagles* | KW | **4** | 2 | **1** | 7.47 | 0.02 |
| | | *Golden eagles* | KW | 2 | 1 | 1 | 5.80 | 0.06 |
| **Impact predators have on lambs which are more than 10 days old** | *0 = NA* | *Foxes* | KW | 1 | 3 | 2 | 2.45 | 0.29 |
| | *1 = None* | | | | | | | |
| | *2 = Low* | | | | | | | |
| | *3 = Medium* | *Badgers* | KW | 0 | 0 | 1 | 4.18 | 0.12 |
| | *4 = High* | *Ravens* | KW | 2 | 2 | 2 | 0.52 | 0.77 |
| | | *Crows* | KW | 2 | 2 | 1 | 4.11 | 0.13 |
| | | *Black-backed gulls* | KW | 1 | 2 | 1 | 0.31 | 0.86 |
| | | *White tailed eagles* | KW | **4** | 3 | **2** | 10.75 | <0.001 |
| | | *Golden eagles* | KW | 2 | 1 | 0 | 7.26 | 0.03 |

2 and 3 agreed that '*reducing blackloss on my farm/croft is important to me*', and all groups agreed that '*understanding the causes of blackloss would help me to reduce lamb losses*'. The study did find a significant difference in responses to the statement that '*the threat to lambs from predators on my holding is low*', with Group 1 strongly disagreeing, and Groups 2 and 3 disagreeing with the statement (KW, $\chi^2(2) = 6.59$, p = 0.04, Table 6).

## Discussion

This questionnaire study set out to capture participants' experiences of blackloss in terms of the causes, consequences, factors and attitudes towards blackloss, as well as examining husbandry practices and lamb health issues within their flocks. Using a typology approach, it was possible to classify respondents into three distinct clusters: Group 1 containing Sheep Stock Clubs (SSC) and very large farms were the largest scale extensive systems, with the biggest size and ewe numbers of the respondent groups, Group 2 consisted of smaller scale crofts and a

**Table 6. Questionnaire items relating to attitudes towards blackloss.** The median response for each of the Groups 1–3 is given.

| Question | Unit | Item | Test | 1 | 2 | 3 | Chi-square | p-value |
|---|---|---|---|---|---|---|---|---|
| **The extent to which you agree or disagree with each of the following statements:** | *0 = Unknown* | *Blackloss is an inevitable part of hill sheep systems.* | KW | 3 | 4 | 4 | 0.12 | 0.94 |
| | *1 = Strongly disagree* | | | | | | | |
| | *2 = Disagree* | *Understanding the causes of blackloss would help me to reduce lamb losses.* | KW | 4 | 4 | 4 | 2.87 | 0.24 |
| | *3 = Neither* | | | | | | | |
| | *4 = Agree* | | | | | | | |
| | *5 = Strongly agree* | *Reducing blackloss on my holding is important to me.* | KW | 5 | 4 | 4 | 5.05 | 0.08 |
| | | *The threat to lambs from predators on my holding is low.* | KW | **1** | **2** | **2** | 6.59 | 0.04 |

SSC, with small hectarage and ewe numbers but which all have access to common grazing, and Group 3 mostly comprised of farmers as well as two crofts, considered to have moderate holding sizes and ewe numbers, but without access to common grazing. This study showed an overall blackloss level on an animal basis of 7.6% between marking and weaning during 2019. It was found that the groups reported similar health challenges, apart from Group 1 which had a significantly higher plochteach challenge. With regards to the role of predators in lamb loss, although the groups had similar assemblages of predators present on their holdings, Group 1 perceived white-tailed eagles as a much higher threat to their lambs than the other groups. Finally, it was observed that many of the respondents were of the opinion that blackloss is inevitable and that predators pose a large threat to lambs, however most agreed that reducing these losses is important and that understanding the causes would enable them to do so.

An overall blackloss level on an animal basis of 7.6% during 2019 was found in this study, slightly higher than the 6.6% reported during 2011 [1]. Nevertheless, the flock level losses of 6.3% (0–17.5%), are well within the levels reported during the earlier study of 6.2% (0–41.4%) [1]. Annual variation in losses is to be expected, due to changes in factors such as weather which affects outdoor lambing conditions and the grazing season etc. A single respondent from Group 1 reported lamb counts for marking and weaning, from which a loss of 1.3% was calculated. This is far below their reported estimated losses of >30%, indicating that losses may be overestimated. However, lamb counts should be treated with some margin of error, as it can be difficult to get a 'perfect gather' where the entire flock is gathered in from the hill, at marking time for example. Group 1 in particular is the most extensive of the groups, and it may be that the lamb count recorded at marking time was far below the actual numbers of lambs on the hill during that time, leading to an under recorded loss. Group 2 had a blackloss level of 4.0%, which was within their estimated loss of 1–10% for 2019. As Group 2 flocks tended to be smaller than either Group 1 or 3 it may be that they are able to gather their flocks more easily to keep accurate records and counts. Blackloss levels were calculated at 13.5% for Group 3, which was above their reported estimated loss of 1–10%, showing the group may be underestimating their losses.

When examining the causes of blackloss, Group 1 felt plochteach/yellowses to be significantly more important than Group 2 did, and they also reported that a greater proportion of their lambs are affected by plochteach in a given year than Group 2. Plochteach tends to be associated with grazing unimproved peatland areas [17, 18], such as those utilised by Group 1 to raise all their lambs. As Group 2 and 3 tended to raise their lambs in semi-improved or improved grazing areas, they may have been less exposed to bog asphodel, the suspected cause of plochteach [18], and had fewer lambs showing the clinical signs of plochteach, therefore considering it to be a less important cause of blackloss in their flocks. Indeed, Groups 2 and 3 both reported that none of their lambs suffer from plochteach in an average year. However, raising lambs on the hill ground is an essential part of hefting, the system by which replacement ewe lambs become accustomed to their home range on the open hill [19–22], and so hill sheep systems face a trade-off between maintaining hefting and potentially exposing their lambs to plochteach photosensitisation.

Group 1 felt that white-tailed eagles presented the highest impact on both their young and older lambs, significantly more-so than Group 3, which viewed them as having no impact on young lambs and a low impact on older lambs, in spite of all Groups having white-tailed eagles present on their holdings and all being located within the range of WTEs. It is interesting that the Groups consistently ranked golden eagles as having a lower impact on lambs than WTEs, despite both species being capable of killing healthy lambs [23–31]. This may be down to the different 'personalities' of the species, with golden eagles being a shier species of the open hills

and much less tolerant of human presence than WTEs, which may make them 'out of sight, out of mind'. WTEs are far more generalist in their habitat requirements, and tend to show tolerance and indeed even curiosity towards humans [29]. This, in combination with their large assuming size (and frequent media coverage), may have led to the higher perceived impact seen here.

When respondent attitudes towards blackloss were examined, there were no significant differences between groups, highlighting that although the groups are themselves significantly distinct, they shared similar attitudes when it comes to blackloss. Although many of the respondents felt that blackloss is inevitable and that predators pose a large threat to lambs, most agreed that reducing these losses is important and that understanding the causes would enable them to do so.

This current study relied on a small sample size of 31 respondents. However, although other typology analysis relied on large sample sizes (e.g. to classify 130 Spanish beef farms based on their size, productivity of labour, degree of specialisation, and degree of extensification [32], and another which grouped 245 Scottish farms based on whether they were business-orientated or environmentally-orientated [33], there have also been some recent British and European studies utilising similar sample sizes to that used here. For example; interviewing 44 hill farms to investigate land management in the Peak District [34], characterisation of 79 terminal sire flocks based on a range of environmental factors [2], clustering 30 Scottish hill farms to examine their responses to policy reforms [3], characterisation of 24 suckler beef farming systems in Scotland [4], categorising the farming practices of 33 French hill farmers [35], and to look at the implementation of a new feeding strategy using two samples of 23 and 79 sheep farmers in Spain [36]. It is likely that this questionnaire would have reached the target audience within the Highlands and Islands, and that sufficient responses of a relatively high quality to undertake a robust typology analysis were attained, successfully characterising respondents into three distinct clusters.

## Conclusions

This study identified three clusters of hill sheep enterprises in the Highlands and Islands of Scotland, namely large, extensive, traditional farms, medium sized farms and crofts, and smallholder farms and crofts. If the clusters identified are representative of the wider issue of blackloss in the Highland and Islands area, this study shows that up to 17.5% of lambs are lost between the marking to weaning period. These are viable lamb losses and are in addition to the industry average loss of 15% which occurs around lambing time [37]. These losses will not be sustainable in the short-term financially, or on long-term due to the hefted nature of hill flocks, not to mention the impact on animal welfare and the toll they take on flockmanager's mental health. The groups faced similar lamb health challenges with regards to parasites (tick borne diseases, fluke and gastrointestinal nematodes), although the large, extensive farms reported higher prevalence of plochteach photosensitisation in their lambs. This group also felt white-tailed eagles posed a higher risk to their lambs than the other groups, despite all groups having similar assemblages of predators. The groups all held similar attitudes and beliefs that blackloss is an inevitable part of hill sheep farming, and that predators pose a risk to their lambs. However most respondents desire to reduce losses, and agree that understanding the causes in greater detail would enable them to do so.

This study has provided information on the suspected causes, consequences and attitudes of blackloss which had not previously been quantified and has gained valuable insight into the often-taboo subject of lamb predation, particularly through suspected white-tailed eagle attacks.

## Supporting information

**S1 File. The General Data Protection Regulation (GDPR) statement and questionnaire which was issued to participants.**
(PDF)

## Acknowledgments

The authors wish to thank all those who published or disseminated information about our questionnaire. We also wish to thank the hill sheep farmers and crofters who responded to our questionnaire. The authors wish to acknowledge Mark Bronsvoort, Margo Chase-Topping and Claire Morgan-Davies for their valuable guidance during the typology analysis. This study is part of a PhD which involves close collaboration between Scotland's Rural College, The Royal (Dick) School of Veterinary Studies, NatureScot and SRUC Veterinary Services.

## Author Contributions

**Conceptualization:** Fiona McAuliffe, Ann McLaren, Neil Sargison, Franz Brülisauer, Andrew Kent, Davy McCracken.

**Data curation:** Fiona McAuliffe.

**Formal analysis:** Fiona McAuliffe.

**Funding acquisition:** Ann McLaren, Neil Sargison, Franz Brülisauer, Andrew Kent, Davy McCracken.

**Investigation:** Fiona McAuliffe.

**Methodology:** Fiona McAuliffe, Ann McLaren, Neil Sargison, Franz Brülisauer, Andrew Kent, Davy McCracken.

**Project administration:** Ann McLaren, Neil Sargison, Franz Brülisauer, Andrew Kent, Davy McCracken.

**Supervision:** Ann McLaren, Neil Sargison, Franz Brülisauer, Andrew Kent, Davy McCracken.

**Visualization:** Fiona McAuliffe.

**Writing – original draft:** Fiona McAuliffe.

**Writing – review & editing:** Fiona McAuliffe, Ann McLaren, Neil Sargison, Franz Brülisauer, Andrew Kent, Davy McCracken.

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
