## [Decision Letter · Decision Letter 0]

29 Nov 2023

PONE-D-23-31486A survey of hill sheep farmer and crofter’s experiences of blackloss in the Highlands and Islands of ScotlandPLOS ONE

Dear Dr. McAuliffe,

Thank you for submitting your manuscript to PLOS ONE. After careful consideration, we feel that it has merit but does not fully meet PLOS ONE’s publication criteria as it currently stands. Therefore, we invite you to submit a revised version of the manuscript that addresses the points raised during the review process.

We look forward to receiving your revised manuscript.

Kind regards,

Julio Cesar de Souza, Ph.D.

Academic Editor

PLOS ONE

Journal Requirements:

"This study was undertaken as part of a PhD studentship, funded by Scotland’s Rural College (SRUC) and NatureScot and registered at the University of Edinburgh’s Royal (Dick) School of Veterinary Studies. It involves close collaboration between SRUC, The Royal (Dick) School of Veterinary Studies, NatureScot and SRUC Veterinary Services." 

"The authors wish to thank all those who published or disseminated information about our 

questionnaire. We also wish to thank the hill sheep farmers and crofters who responded to our 

questionnaire. The authors wish to acknowledge Mark Bronsvoort, Margo Chase-Topping and Claire 

Morgan-Davies for their valuable guidance during the typology analysis. This study is part of a PhD 

funded by SRUC and NatureScot and registered at the University of Edinburgh’s Royal (Dick) School of 

Veterinary Studies. It involves close collaboration between SRUC, The Royal (Dick) School of Veterinary 

Studies, NatureScot and SRUC Veterinary Services."

"This study was undertaken as part of a PhD studentship, funded by Scotland’s Rural College (SRUC) and NatureScot and registered at the University of Edinburgh’s Royal (Dick) School of Veterinary Studies. It involves close collaboration between SRUC, The Royal (Dick) School of Veterinary Studies, NatureScot and SRUC Veterinary Services." 

6. Please include a copy of Table 7 which you refer to in your text on page 13 in PDF submission.

**Additional Editor Comments:**

Dear Sir,

considering the opinions of the reviewers for this paper,

we ask you to consider the suggestions so that we can move forward with the publication.

Best Regards,

Julio Souza

Reviewers' comments:

Reviewer's Responses to Questions

**Comments to the Author**

1. Is the manuscript technically sound, and do the data support the conclusions?

Reviewer #1: Yes

Reviewer #2: Yes

2. Has the statistical analysis been performed appropriately and rigorously? 

Reviewer #1: Yes

Reviewer #2: Yes

3. Have the authors made all data underlying the findings in their manuscript fully available?

Reviewer #1: Yes

Reviewer #2: Yes

4. Is the manuscript presented in an intelligible fashion and written in standard English?

Reviewer #1: Yes

Reviewer #2: Yes

5. Review Comments to the Author

Reviewer #1: General comment: Well written article, has viable information about topic of this manuscript. The abstract is covered this manuscript completely. It contains very long sentences. I suggest that it be rephrased in short sentences. Results: good and clear. I suggest that the authors indicate their findings in detail, especially the results of tables 1 and 4. Conclusion: Authors wrote the conclusion of their study in abbreviated. I suggest that they rewrite the conclusion in detail, based on the results of their study.

References: excellent, clear and complete and important references.

- Some references are old such as: 1963, 1964, 1967, 1971, 1986, 1987, 1990. I suggest replace it with modern references.

Reviewer #2: The paper present a well planned and carried out survey with some interesting results that could be relevant for ameliorate the mortality rate of lambs in extensive productive system in Scotland.

I only have found a sentence that have no meaning for me, and I recommend to be reword or explain in a different way and is in Lines 304 to 306: “A single respondent from Group 1 reported lamb counts for marking and weaning, with a loss of 1.3%. This is far below the reported estimated losses of 11-20% for Group 1 on a whole, and the >30% loss reported by this respondent, indicating that losses may be overestimated.” It is not clear for me if the same respondent of Group 1 reported lamb losses of 1.3 % and the same respondent reported more that 30% loss. This those do not make sense to me.

6. PLOS authors have the option to publish the peer review history of their article (what does this mean?). If published, this will include your full peer review and any attached files.

Reviewer #1: No

Reviewer #2: **Yes: **Marta E. Alonso

---

## [Author Response · Author response to Decision Letter 0]

9 Jan 2024

Reviewer 1:

General comment: Well written article, has viable information about topic of this manuscript.

Title of this manuscript is excellent. 

The abstract is covered this manuscript completely. It contains very long sentences. I suggest that it be rephrased in short sentences.

The abstract has been edited to break sentences into shorter sentences to improve clarity.

Introduction: readable, comprehensive, and covering the subject quite right. It is consist of an important references. 

Materials and methods: perfect.

Results: good and clear. I suggest that the authors indicate their findings in detail, especially the results of tables 1 and 4. 

Further detail has been added when presenting the results from Tables 1 and 4.

Discussion: excellent and clear. 

Conclusion: Authors wrote the conclusion of their study in abbreviated. I suggest that they rewrite the conclusion in detail, based on the results of their study.

Further detail has been added to the conclusion section to reiterate the key findings and results of the study.

References: excellent, clear and complete and important references.

- Some references are old such as: 1963, 1964, 1967, 1971, 1986, 1987, 1990. I suggest replace it with modern references. 

Some additional modern references have been added (Fisher and Matthews, 2001). Others are seminal works are should be referenced. See below for explanations on each reference.

Tables: excellent and clear.

Figures: excellent and clear.

Reviewer 2:

The paper present a well planned and carried out survey with some interesting results that could be relevant for ameliorate the mortality rate of lambs in extensive productive system in Scotland.

I only have found a sentence that have no meaning for me, and I recommend to be reword or explain in a different way and is in Lines 304 to 306: “A single respondent from Group 1 reported lamb counts for marking and weaning, with a loss of 1.3%. This is far below the reported estimated losses of 11-20% for Group 1 on a whole, and the >30% loss reported by this respondent, indicating that losses may be overestimated.” It is not clear for me if the same respondent of Group 1 reported lamb losses of 1.3 % and the same respondent reported more that 30% loss. This those do not make sense to me.

This sentence has been reworded to improve clarity. From the lamb counts the study respondent returned we were able to calculate a loss of 1.3%, however the respondent themselves estimated their losses to be >30%.

References:

5 Kaufman L, Rousseeuw PJ. Partitioning Around Medoids (Program PAM). In: Kaufman L, Rousseeuw PJ, editors. Finding Groups in Data: An Introduction to Cluster Analysis. Hoboken, NJ, USA: John Wiley & Sons, Inc.; 1990. pp. 68–125. doi:10.1002/9780470316801

The is the seminal first paper to describe partitioning around medoids, which is the clustering method used in this study, and should therefore be reference.

6 MacQueen J. Classification and analysis of multivariate observations. Fifth Berkeley Symposium on Mathematical Statistics and Probability. Los Angeles LA USA: University of California.; 1967. pp. 281–297.

This is the seminal paper describing k-means clustering, an alternative clustering method, and should therefore be included in the references.

14 Gower JC. A General Coefficient of Similarity and Some of Its Properties. Int Biometric Soc. 1971;27: 857–871. doi:10.2307/2528823

This is the seminal work which described Gower’s distance matrix, which is what was used in the current analysis, and so this paper muct be referenced.

20 Hunter RF, Milner C. The behaviour of individual, related and groups of South Country Cheviot hill sheep. Anim Behav. 1963;11: 507–513. doi:10.1016/0003-3472(63)90270-7

21 Lawrence AB. Mother-daughter and peer relationships of Scottish hill sheep. Anim Behav. 1990;39: 481–486. doi:10.1016/S0003-3472(05)80412-9

The 1963 and 1990 papers descibe sheep home range behaviour in a Scottish context, relvant to the current study and will therefore be retained. Although they are older references they are accompanied by more modern references from 2001 and 2016.

22 Lockie JD. The breeding density of the golden eagle and fox in relation to food supply in Wester Ross, Scotland. Scottish Nat. 1964;71: 67-77.

23 Leitch AF. Report on eagle predation on lambs in the Glenelg area in 1986. 1986.

24 Matchett MR, O’Gara BW. Methods of controlling golden eagle depredation on domestic sheep in southwestern Montana. J Raptor Res. 1987;21: 85–94.

We feel that these references, particularly Lockie (1964) and Leitch (1986) were very important as some of the earliest investigations into eagle depredation of lambs within a Scottish context. Matchett (1987) provides examples of eagle predation of lambs from Montana, USA, however as the sheep husbandry practices are similar to Scottish hill farming it provides a good comparison, and is worthy of inclusion. Several other modern references (2004, 2010, 2013, 2015, and 2021) with relation to eagle predation are also provided.

---

## [Editor Report · Decision Letter 1]

23 Jan 2024

A survey of hill sheep farmer and crofter’s experiences of blackloss in the Highlands and Islands of Scotland

PONE-D-23-31486R1

Dear Dr. McAuliffe,

We’re pleased to inform you that your manuscript has been judged scientifically suitable for publication and will be formally accepted for publication once it meets all outstanding technical requirements.

Kind regards,

Julio Cesar de Souza, Ph.D.

Academic Editor

PLOS ONE

Additional Editor Comments (optional):

Dear authors,

thanks to adjust.

The paper is accepted.

Best regards

Julio Souza
---

## [Editor Report · Acceptance letter]

18 Mar 2024

PONE-D-23-31486R1 

PLOS ONE

Dear Dr. McAuliffe, 

I'm pleased to inform you that your manuscript has been deemed suitable for publication in PLOS ONE. Congratulations! Your manuscript is now being handed over to our production team.

Kind regards, 

on behalf of

Dr. Julio Cesar de Souza 

Academic Editor

PLOS ONE